# Easing Training Process of Rectified Flow Models Via Lengthening Inter-Path Distance

**Shifeng Xu**[1]         **Yanzhu Liu**[2]         **Adams Wai-Kin Kong**[1]

[1]College of Computing and Data Science, Nanyang Technological University, Singapore
shifeng001@e.ntu.edu.sg,        adamskong@ntu.edu.sg
[2]Institute for Infocomm Research (I2R) & Centre for Frontier AI Research, A*STAR, Singapore
liu_yanzhu@i2r.a-star.edu.sg

## Abstract

Recent research pinpoints that different diffusion methods and architectures trained on the same dataset produce similar results for the same input noise. This property suggests that they have some preferable noises for a given sample. By visualizing the noise-sample pairs of rectified flow models and stable diffusion models in two-dimensional spaces, we observe that the preferable paths, connecting preferable noises to the corresponding samples, are better organized with significant fewer crossings compared with the random paths, connecting random noises to training samples. In high-dimensional space, paths rarely intersect. The path crossings in two-dimensional spaces indicate a shorter inter-path distance in the corresponding high-dimensional spaces. Inspired by this observation, we propose the Distance-Aware Noise-Sample Matching (DANSM) method to lengthen the inter-path distance for speeding up the model training. DANSM is derived from rectified flow models, which allow using a closed-form formula to calculate the inter-path distance. To further simplify the optimization, we derive the relationship between inter-path distance and path length, and use the latter in the optimization surrogate. DANSM is evaluated on both image and latent spaces by rectified flow models and diffusion models. The experimental results show that DANSM can significantly improve the training speed by 30% ∼ 40% without sacrificing the generation quality. Code: https://github.com/shifengxu/DANSM.

## 1 Introduction

Diffusion-based generative models, such as diffusion models (Ho et al., 2020; Nichol & Dhariwal, 2021; Song et al., 2021a; Dhariwal & Nichol, 2021; Rombach et al., 2022) and rectified flow models (Liu et al., 2023a;b; Liu, 2022; Lipman et al., 2022), have garnered considerable attention due to their high-quality generation and broad range of applications. Training diffusion-based generative models is in fact establishing a mapping between the noise space and sample space. Recent research discovers that when training on the same dataset, different diffusion methods using different architectures result in a similar mapping. In other words, given the same input noise, the trained models generate similar resultant samples. Fig. 1 demonstrates such mappings. This phenomenon,

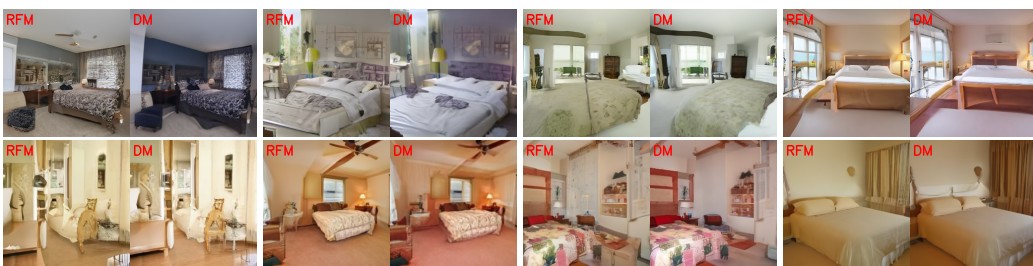

Figure 1: Images generated by well-trained rectified flow model (RFM) (Liu et al., 2022) and diffusion model (DM) (Ermongroup, 2021). Images in the same group are derived from the same noise.

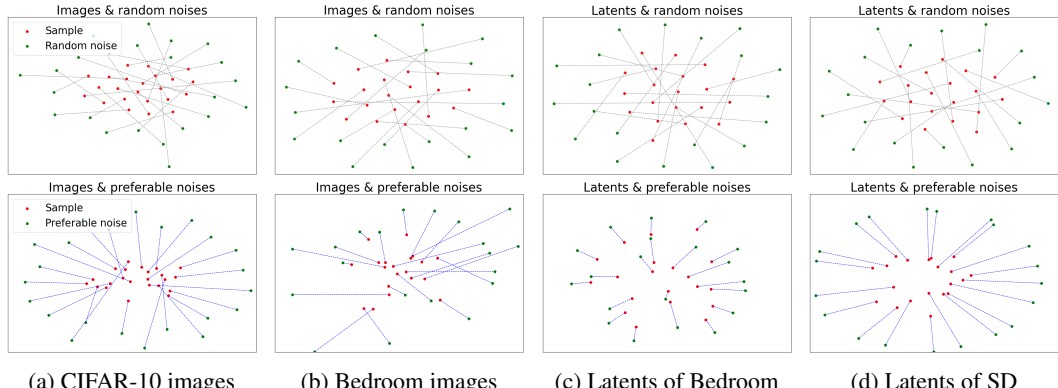

Figure 2: Visualization of samples and noises using t-SNE (Van der Maaten & Hinton, 2008). Each sample and its corresponding noise are connected with dashed line. The lines in the top row are messy with multiple intersections, while the bottom row illustrates well-organized lines.

known as "consistent model reproducibility" in Zhang et al. (2024), has been discussed in some prior works (Song et al., 2021b; Liu et al., 2023a). They attempt to understand this phenomenon from the denoising score matching perspective (Vincent, 2011).

The consistent model reproducibility phenomenon suggests that given a clean or generated sample, there exists some preferable noises, from which the well-trained models can generate the sample. Fig. 2 visualizes the clean samples, preferable noises, and random noises through t-SNE (Van der Maaten & Hinton, 2008). The preferable noises (the green dots in the second row) are obtained from well-trained rectified flow models and stable diffusion model through a sample-to-noise process. The gray lines in the first row represent the random paths connecting the random noise and clean samples of training process, and the blue lines in the second row denote the preferable paths linking preferable noises and their corresponding samples. Fig. 2 shows clearly that the random paths used in model training are very messy with many intersections, but the preferable paths obtained from fully training models are well-organized. We argue that the complex random path patterns hamper the training process and significantly slow it down. This aligns the analysis in Li et al. (2024). Notice that in a high-dimensional space, intersections of two path are very rare. The intersections in the two-dimensional visualization imply that the paths are closer in those regions in the original high dimensional space. Inspired by the difference between the messy random path pattern used in training process and the well-structured preferable path pattern from fully trained model, we propose the Distance-Aware Noise-Sample Matching (DANSM) method to improve training speed. In this work, we concentrate on rectified flow models in DANSM derivation, because its paths are straight-line segments, which provide an effective closed-form solution for the inter-path distance calculation. We further deduce the negative correlation between inter-path distance and path length, and use it in the DANSM optimization surrogate. Although DANSM is derived from rectified flow models, we also test it on diffusion models, because of their popularity. Extensive experiments are conducted on image and latent spaces of rectified flow models and diffusion models. The results show that DANSM significantly enhances the training process up to 30%∼40%.

The contributions of this work are summarized as follows:

- Inspired by the difference between random paths used in training and preferable paths from well-trained models, we propose DANSM method aiming to increase the inter-path distance and enhance the training process.

- Based on the straight paths of rectified flow models, we first derive a closed-from formula to calculate inter-path distance. Furthermore, we prove the negative correlation between inter-path distance and path length, and use the latter in the optimization surrogate of the DANSM method.

- Extensive experiments are conducted on image and latent spaces of rectified flow models and diffusion models, which demonstrate DANSM's outstanding capabilities for speeding up the training process.

## 2 PRELIMINARIES AND RELATED WORKS

Without loss of generality, let $x$ be a data sample from distribution $\pi_0$, $z$ be a noise from standard Gaussian noise $\mathcal{N}(\mathbf{0}, \mathbf{I})$, and $t \in [0, 1]$ be the timestep. $x$ and $z$ can be transformed into each other with $t$. This paper assumes a continuous transformation between $x$ and $z$, $x^{(t)} := x(t)$, where $x^{(0)} = x$ and $x^{(1)} = z$.

### 2.1 RECTIFIED FLOW MODELS

**Rectified Flow Models** (RFM) utilize Ordinary Differential Equations (ODE) to approximate the straight paths between noises and samples. They offer a unique solution to generative modeling from the perspective of optimal transport (Villani, 2009; 2021). In RFM, each point on the noise-sample path is a linear interpolation between them.

$$x^{(t)} = (1 - t)x + tz, \quad \frac{dx^{(t)}}{dt} = z - x, \quad t \in [0, 1]. \tag{1}$$

It is worth noting that the expression of $x^{(t)}$ in Eq. 1 is straightforward and its gradient remains stable. In fact, Esser et al. (2024) also mentioned that RFM provides better theoretical properties and conceptual simplicity compared to classic diffusion models (Song et al., 2021a; Ho et al., 2020). When training RFM, the model $\theta$ is expected to drive the flow to follow the interpolation between $x$ and $z$. It can be achieved by solving a simple least squares regression problem (Liu et al., 2023a):

$$\mathcal{L}_{RFM}(x, z, t; \theta_{RFM}) = \mathbb{E}_{x \sim \pi_0, z \sim \mathcal{N}(\mathbf{0}, \mathbf{I}), t \sim U(0,1)} \big[ \|(z - x) - \theta_{RFM}(x^{(t)}, t)\|_2^2 \big], \tag{2}$$

where $U(0, 1)$ is the uniform distribution between 0 and 1.

In Eq. 2, the sample $x$ and noise $z$ are randomly paired for loss calculation. However, as mentioned in the introduction, well-trained RFM have some preferable noises for a given sample. This implies that in most cases, the sample $x$ is not paired up with its preferable noise, which can significantly hinder the model training process. This paper aims to explore this issue and propose an optimization solution to address it.

### 2.2 CONSISTENT MODEL REPRODUCIBILITY

In the realm of diffusion-based generative modeling, the consistent model reproducibility phenomenon has been discussed by several works. The concept of "uniquely identifiable encoding", as discussed in Song et al. (2021b) (page 7), suggests that the encoding (sample) for an input (noise) is uniquely determined by the data distribution. Zhang et al. (2024) introduce the concept of "consistent model reproducibility" and state that when trained on the same dataset, various diffusion-based models generate similar data samples. However, they did not utilize it to ease the model training process.

Theoretically, the consistent model reproducibility phenomenon can be understood through the lens of denoising score matching (Vincent, 2011), as interpreted via Tweedie's formula. Here, $x^{(0)}$ is a clean sample with no noise, and $x^{(t)}$ is a noisy variable with noise level $\sigma_t$. The objective of model training is to estimate $\mathbb{E}[x^{(0)}|x^{(t)}]$, which represents the mean of all clean samples that could produce $x^{(t)}$ when perturbed by noise of level $\sigma_t$. Since the clean samples used for training are fixed, the mapping between $x^{(t)}$ and $x^{(0)}$ should remain consistent. We would like to highlight that the aim of this paper is not to analyze consistent model reproducibility. However, it inspires us to maximize the inter-path distance to speed up the training.

## 3 METHOD

In this section, we first provide a closed-form formula to calculate the inter-path distance for RFM. Next, we explain how inter-path distance influences the quality of training data and the overall training process. Finally, we introduce the DANSM method, which lengthens the inter-path distance, thereby accelerating the model training process.

### 3.1 INTER-PATH DISTANCE

In RFM, the path between noise and sample is represented by a linear interpolation between them. When calculating the inter-path distance, the timestep $t$ should be taken into account, as it determines the proportion of noise and sample at each point along the interpolation. Therefore, the distance between two paths can be considered as a function of $t$. Given two noise-sample pairs, $(z_1, x_1)$ and $(z_2, x_2)$, where $z_1$ and $z_2$ are noises and $x_1$ and $x_2$ are samples, the paths are defined with timestep $t$ as follows:

$$\begin{cases} r_1 = (1-t)x_1 + tz_1 \\ r_2 = (1-t)x_2 + tz_2 \end{cases} \qquad t \in [0,1]. \tag{3}$$

Let $V = x_2 - x_1$ and $U = z_2 - z_1$. The distance between $r_1$ and $r_2$ is an interpolation of $V$ and $U$: $f_{r_1,r_2}(t) = \|(1-t)V + tU\|_2$. Since $t \in [0,1]$, the minimal value occurs at timestep $t^*$:

$$t^* = \begin{cases} 0 & \text{if} \quad \hat{t} \le 0 \\ \hat{t} & \text{if} \quad \hat{t} \in [0,1] \\ 1 & \text{otherwise} \end{cases} \quad , \quad \text{where} \quad \hat{t} = \frac{V^\top(V-U)}{(V-U)^\top(V-U)}. \tag{4}$$

The detailed deduction of Eq. 4 is given in appendix A.1. Computing $\hat{t}$ requires one vector subtraction, two dot products and one division, and calculating $f_{r_1,r_2}(t)$ requires additional one vector addition, two scalar multiplications and one norm computation. Directly using $f_{r_1,r_2}(t)$ to compute all possible inter-path distances of $n$ paths is inefficient due to the complicated operations. Therefore, in this paper, the distance of two paths $w.r.t.$ timestep $t$ is defined as the minimal distance on timestep $t^*$:

$$dist(r_1, r_2) = \min_{t \in [0,1]} f_{r_1,r_2}(t) = f_{r_1,r_2}(t^*). \tag{5}$$

### 3.2 INTER-PATH DISTANCE MATTERS IN TRAINING PROCESS

In the training process of RFM, the training data are actually the points in the noise-sample paths. The quality of these paths is significantly influenced by the inter-path distances. To elucidate this, we consider an extreme case on RFM models. Consider two paths, $P_1 = (z_1, x_1)$ and $P_2 = (z_2, x_2)$, that intersect at a specific timestep $t^*$. At $t^*$, the state of the noisy sample satisfies such equation:

$$x^{(t^*)} = (1-t^*)x_1 + t^*z_1 = (1-t^*)x_2 + t^*z_2. \tag{6}$$

However, when training the model $\theta$ on $x^{(t^*)}$ and $t^*$, the expected outputs differ based on the paths being followed as shown in Eq. 7. This discrepancy confuses the model training, as different paths lead to different gradients. To avoid such conditions, sufficient inter-path distance is necessary.

$$\text{Expected model prediction:} \ \theta\big(x^{(t^*)}, t^*\big) = \begin{cases} z_1 - x_1 & \text{if training on path } P_1 \\ z_2 - x_2 & \text{if training on path } P_2 \end{cases}. \tag{7}$$

To analyse how the inter-path distances impact model training, we delve deeper into the inter-path distances. For better elaboration, we define the terms as below.

- *preferable path*: The path between a sample and its preferable noise, which is obtained by sample-to-noise process (the inverted sampling process) on fully trained model.
- *random path*: The path between a sample and any random noise.
- *average distance*: For $n$ paths, there are $c = n(n-1)/2$ path pairs, resulting in $c$ inter-path distances. The average of these distances is termed the average distance of the $n$ paths.
- *minimal distance*: For $n$ paths, each path has a nearest path, yielding $n$ minimal distances. The mean of those distances is called the minimal distance of the $n$ paths.
- *training loss*: For RFM, the ground-truth gradient is $z - x$ as in Eq. 1, and the predicted gradient is denoted as $\tilde{g}$. The squared $\ell_2$ norm $\|(z-x) - \tilde{g}\|^2$ is called training loss.

As shown in Fig. 4 of Sec. 5.1, when comparing by both average and minimal distances, the preferable paths exhibit greater inter-path distances than random paths. Additionally, as the inter-path distances increase, the training losses decrease, indicating an improvement in the quality of noise-sample paths. This aligns with our argument that shorter inter-path distances imply lower path quality, thereby making the training process difficult.

### 3.3 DISTANCE-AWARE NOISE-SAMPLE MATCHING

DANSM's problem setting is defined as follows. Given a set of $n$ sample points and a set of $n$ noise points, the goal is to establish a one-to-one mapping between the two sets, resulting in $n$ paths. Let $p_{i,j}$ be the path from the $i$-th noise point to the $j$-th sample point. The objective of DANSM is to maximize the inter-path distances of those paths while ensuring the process can be completed within in a reasonable amount of time. In this paper, the sample (or noise) set size $n$ is referred as *match-size*, which serves as a key parameter. The objective is defined as:

$$\max_{\sigma} \frac{2}{n(n-1)} \sum_{i=1}^{n} \sum_{j=i+1}^{n} dist(p_{i,\sigma(i)}, p_{j,\sigma(j)}), \tag{8}$$

where $\sigma$ is a permutation with input from $1, 2, \cdots, n$, and $dist(p_{i,\sigma(i)}, p_{j,\sigma(j)})$ is the distance between the paths $p_{i,\sigma(i)}$ and $p_{j,\sigma(j)}$, as specified in Eq. 5. It is worth noting that the noise set has the same size as the sample set, ensuring that all noises are utilized in the training process without any being discarded. This is crucial because if the noise set contains more points than the sample set, certain noises will inevitably be excluded from the model training. Due to the relationship between inter-path distance and path length, the unused noises are those that are farthest from the samples, meaning certain regions in the noise space remain untrained. This lack of training in specific noise space areas can negatively impact the overall generation performance.

Although the objective of the DANSM method is clear, calculating the inter-path distance for multiple paths in high-dimensional space is computationally expensive, especially for a large number of paths. To address this issue, we identified a surrogate approach to manipulate the inter-path distance without calculating it. This approach is elaborated in Sec. 4.

## 4 OPTIMIZATION SURROGATE

To further analyze the inter-path distance, we introduce the path length, which is the length of the path between noise and sample. In this section, we prove the negative correlation between inter-path distance and path length. Building on this insight, we propose an optimization surrogate of DANSM, which aims to shorten the path lengths, thereby increasing the inter-path distances accordingly.

### 4.1 NEGATIVE CORRELATION BETWEEN INTER-PATH DISTANCE AND PATH LENGTH

Note that the noises and samples of RFM have the same dimension and therefore, they can be considered from the same high-dimensional space $\mathbb{R}^d$, which encompasses both noise space and sample space. We elaborate the correlation between inter-path distance and path length in $\mathbb{R}^d$. The path of RFM is the line segment between noise point and sample point, while the inter-path distance is defined in Sec. 3.1.

We begin with a simple scenario containing only two paths. Let the sample points be $x_1$ and $x_2$, and the noise points be $z_1$ and $z_2$. Consequently, the paths between samples and noises have two cases: $P_1 = (z_1, x_1)$, $P_2 = (z_2, x_2)$ and $Q_1 = (z_1, x_2)$, $Q_2 = (z_2, x_1)$. Following Sec. 3.1, we define the vectors $U = z_2 - z_1$, $V = x_2 - x_1$ and $\overline{V} = x_1 - x_2$. Notably, $\overline{V} = -V$ and $\|V\| = \|\overline{V}\|$, where $\|\cdot\|$ means the length of a vector or line segment. These variables are illustrated in Figs. 3a and 3b.

As shown in Sec. 3.1, the inter-path distance between $P_1$ and $P_2$ is an interpolation of $U$ and $V$. Similarly, the inter-path distance between $Q_1$ and $Q_2$ is an interpolation of $U$ and $\overline{V}$. These relationships are shown in Figs. 3c and 3d. Let $\angle ecf = \gamma$ and $\angle gcf = \overline{\gamma}$. Since $V = -\overline{V}$, $\gamma$ and $\overline{\gamma}$ are supplementary angles for each other. Using the law of cosine, we derive:

$$cos\, \gamma = \frac{\|U\|^2 + \|V\|^2 - \|U-V\|^2}{2\|U\| \cdot \|V\|} = \frac{\|q_1\|^2 + \|q_2\|^2 - (\|p_1\|^2 + \|p_2\|^2)}{2\|U\| \cdot \|V\|}, \tag{9}$$

where $p_1 = x_1 - z_1$, $p_2 = x_2 - z_2$, $q_1 = x_2 - z_1$ and $q_2 = x_1 - z_2$. The detailed derivation of Eq. 9 is provided in the appendix A.2. In the same way, we have:

$$cos\, \overline{\gamma} = \frac{\|U\|^2 + \|\overline{V}\|^2 - \|U-\overline{V}\|^2}{2\|U\| \cdot \|\overline{V}\|} = \frac{\|p_1\|^2 + \|p_2\|^2 - (\|q_1\|^2 + \|q_2\|^2)}{2\|U\| \cdot \|\overline{V}\|} = -cos\, \gamma. \tag{10}$$

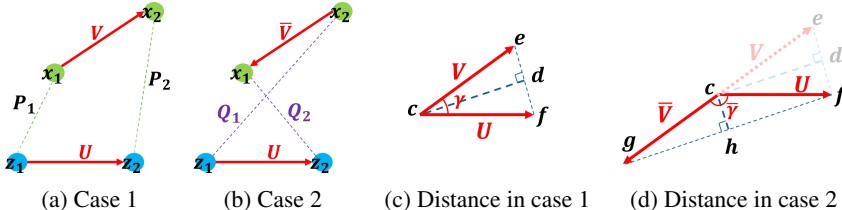

|  (a) Case 1 | (b) Case 2 | (c) Distance in case 1 | (d) Distance in case 2 |

Figure 3: Inter-path distances in a simple scenario of two sample points and two noise points.

---

**Algorithm 1** Surrogate method of DANSM

---

**Input:** a noise set $\mathcal{Z}$ of size $n$, and a sample set $\mathcal{X}$ of size $n$.
**Output:** a noise-sample path set $\mathcal{P}$ of size $n$.
1: Initialize path set $\mathcal{P} = \phi$
2: **for** each sample $x$ in $\mathcal{X}$ **do**
3:     From $\mathcal{Z}$, find $z$ who has shortest path length with $x$        ▷ search all in $\mathcal{Z}$
4:     Remove noise $z$ from $\mathcal{Z}$                                    ▷ each noise is used only once
5:     Add path $p(z, x)$ into $\mathcal{P}$
6: **end for**

---

This reveals the relationship between the path lengths and the inter-path distances. For case 1, as shown in Figs. 3a and 3c, the inter-path distance between $P_1$ and $P_2$ is $\|c - d\|$, where $\overline{cd} \perp \overline{ef}$ and $d$ is the foot of the perpendicular. Similarly for case 2, as illustrated in Figs. 3b and 3d, the inter-path distance of $Q_1$ and $Q_2$ is $\|c - h\|$. Assume that $\|p_1\|^2 + \|p_2\|^2 < \|q_1\|^2 + \|q_2\|^2$, we have $cos\ \gamma > 0$ and $cos\ \overline{\gamma} < 0$, indicating that $\|c - d\| > \|c - h\|$. This conclusion is proved in appendix A.3.

Based on the above deduction, we can conclude that between two samples and two noises, the paths with shorter lengths result in longer inter-path distance, while the paths with longer lengths lead to shorter inter-path distance.

## 4.2 SURROGATE METHOD OF DANSM

As proved in Sec. 4.1, the path length is negatively correlated with the inter-path distance, indicating that decreasing the former will increase the latter accordingly. This relationship can be utilized in the surrogate method of DANSM. When assigning noises to samples, we aim to shorten the noise-sample path lengths, which in turn lengthen the inter-path distances. The minimization objective is defined in Eq. 11, with a lower computational complexity than the maximization objective in Eq. 8.

$$\min_{\sigma} \frac{1}{n} \sum_{i=1}^{n} \|p_{i,\sigma(i)}\|. \tag{11}$$

To further reduce processing time, a greedy algorithm is employed to pair up noise and sample. Specifically, for each sample in $\mathcal{X}$, it selects the nearest available noise from $\mathcal{Z}$ for pairing. The detailed steps are shown in Alg. 1. In this way, we do not need to calculate the exact inter-path distance and significantly reduce the computation time. This surrogate method is simple, straightforward, and most importantly, fast. Its Euclidean distance calculation is highly suitable for GPU parallelization, allowing it to be executed efficiently.

A recent work, Immiscible Diffusion (Li et al., 2024), employs a similar approach to our surrogate method for assigning noises to samples. It aligns with the goal of our DANSM method, but its algorithm functions as another surrogate of DANSM and has not been evaluated on RFM. It is grounded in physics intuition and analogy, lacking a rigorous mathematical foundation. While its noise and image batches have the same size, it does not provide a clear rationale that why "equal-sized" batches are necessary. It focuses on shortening noise-sample path lengths but fails to establish the negative correlation between path length and inter-path distance. In summary, although Immiscible Diffusion identifies an effective method, it does not offer deeper explanations for its findings. Further comparison is given in Sec. 5.6.

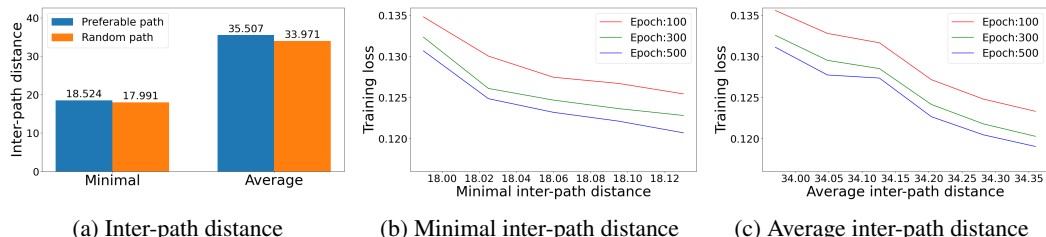

(a) Inter-path distance    (b) Minimal inter-path distance    (c) Average inter-path distance

Figure 4: Inter-path distances of CIFAR-10 data from well-trained RFM. (a) Minimal and average inter-path distances of the preferable and random paths. (b),(c): Training losses vs. distances.

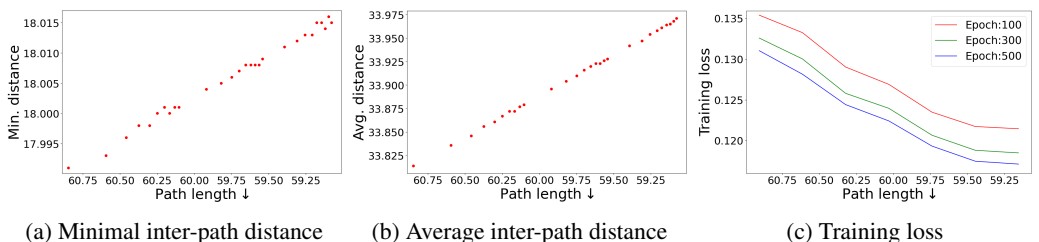

(a) Minimal inter-path distance    (b) Average inter-path distance    (c) Training loss

Figure 5: Path length analysis from well-trained RFM on CIFAR-10. (a), (b): Relationship between path length and inter-path distances. (c): Relationship between path length and training loss.

## 5 EXPERIMENTS

To verify the effectiveness of the DANSM method on RFM, experiments are conducted on both image space and latent space. For image space, three datasets CIFAR-10 (Krizhevsky et al., 2009), ImageNet64 (Deng et al., 2009), and LSUN Bedroom (Yu et al., 2015) are tested. They have image size of 32×32, 64×64, and 256×256, respectively. For latent space, the autoencoder of Stable Diffusion (SD) (Rombach et al., 2022) is utilized for the encoding and decoding between images and latent variables. The latent variables serve as clean samples in the training and sampling processes. Once sampling is completed, the generated latent variables are decoded into real images for evaluation. Throughout the experiments, the noises $z \sim \mathcal{N}(\mathbf{0}, \mathbf{I})$, the image pixel values are normalized to the range of $[-1, 1]$, and the latent variables are kept as their original values without scaling.

### 5.1 INTER-PATH DISTANCE MATTERS IN TRAINING PROCESS

In this paper, we propose to ease the training of rectified flow models by lengthening the inter-path distance. Therefore, when conducting the experiments, we firstly validate the importance of inter-path distance in the training process. We compare the inter-path distance of preferable path and random path in Fig. 4a. The definitions of preferable and random paths can be found in Sec. 3.2. Based on the well-trained RFM (Liu et al., 2022) for CIFAR-10, the average inter-path distance of preferable paths is 35.5, which is greater than that of random paths — 33.9. Similar results are disclosed for minimal inter-path distance. Moreover, Figs. 4b and 4c depict how training loss evolves as the inter-path distance changes: with distance increasing, the training loss of RFM model decreases accordingly. These results offer the basis that inter-path distance plays a critical role in reducing training loss and improving the model training process.

### 5.2 NEGATIVE CORRELATION BETWEEN INTER-PATH DISTANCE AND PATH LENGTH

As discussed in Sec. 4.1, path length is negatively correlated with inter-path distance, which is why we can use path length as the optimization surrogate. In this section, we validate this relationship through experiments with RFM on CIFAR-10 dataset. As shown in Fig. 5, when path lengths decrease, the related minimal and average inter-path distances increase, and the training loss decreases accordingly. The decreasing training losses shown in Fig. 5c highlight that shortening the path lengths improves the training data quality, which can, in turn, ease the model's training process.

Table 1: FID↓ comparison during RFM training at different epochs and DANSM match-sizes ("*ms*").

| | CIFAR-10 (10-step sampling) | | | | | | ImageNet64 (5-step sampling) | | | | | | Bedroom (10-step sampling) | | |
|---|---|---|---|---|---|---|---|---|---|---|---|---|---|---|---|
| *epoch* | 100 | 200 | 300 | 400 | 500 | *epoch* | 40 | 80 | 120 | 160 | 200 | *epoch* | 20 | 40 | 60 |
| baseline | 24.1 | 19.9 | 18.2 | 17.2 | 16.5 | baseline | 74.2 | 68.4 | 64.3 | 60.8 | 57.1 | baseline | 84.3 | 68.0 | 34.7 |
| ms=1,000 | 20.9 | 16.6 | 14.8 | 14.2 | 13.5 | ms=100 | 74.1 | 67.5 | 61.2 | 57.6 | 54.7 | ms=100 | 69.6 | 67.3 | 28.9 |
| ms=10,000 | **17.5** | **12.7** | **11.1** | **10.3** | **9.7** | ms=1,000 | **72.2** | **64.5** | **58.1** | **54.5** | **52.2** | ms=500 | **40.3** | **27.6** | **25.2** |

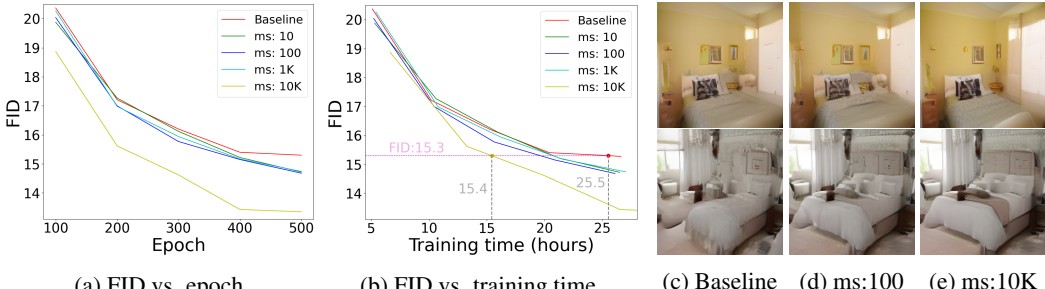

(a) FID vs. epoch  (b) FID vs. training time  (c) Baseline  (d) ms:100  (e) ms:10K

Figure 6: Comparisons between baseline RFM and DANSM with various match-size("*ms*") on latent space of LSUN dataset by 10-step sampling. (c)-(e) Example images generated by different training models at the same training time (15.4 hours). A.5 includes more comparisons.

## 5.3  THE PERFORMANCE OF DANSM IN IMAGE SPACE

When training in image space, the DANSM method exhibits superior performance in RFM models compared to the vanilla baseline. To quantify this performance improvement, we evaluate the models at various stages during the training process. At specific epochs, when the model has not yet fully converged, we generate samples using the model and compute FID scores to compare the effectiveness of DANSM against the baseline. Tab. 1 reports FID scores across different epochs (in columns) and match-sizes (in rows) on different datasets. The last two rows of the table are DANSM models saved out on the specific epochs. The DANSM method consistently achieves lower FID scores compared to the baseline, with the lowest FID scores highlighted in bold font. On CIFAR-10, DANSM with match-size 10,000 yields best performance, reducing the FID from 16.5 to 9.7. Similarly, on Bedroom dataset, the DANSM method achieves lower FID scores than the baseline method, demonstrating its effectiveness on large images.

## 5.4  THE PERFORMANCE OF DANSM IN LATENT SPACE

We evaluate the DANSM method in the latent space by using autoencoder of SD for image synthesis. In the experiments, the dimension of latent space is $4\times32\times32$, and the latent variables are encoded from 50,000 LSUN Bedroom images with the resolution of $256\times256$. To evaluate the effectiveness of the DANSM method, we train the models from scratch with different match-sizes. At specific epochs during the training process, latent samples are generated by the model and decoded into real images by the decoder of SD to calculate FID scores. Additionally, we track the training time to evaluate the improvement in training efficiency. The FID scores over epoch and training time are given in Fig. 6. As shown in the figure, DANSM achieves same FID in significantly less number of epochs and shorter training time for RFM, where match-size of 10K has the best performance.

## 5.5  TRAINING PROCESS ACCELERATION

The DANSM method aims to lengthen the inter-path distance, which eases the training process and reduces training time. However, DANSM itself introduces computational overhead as well, which needs to be carefully managed. A critical consideration is whether the performance gained from DANSM can justify the additional computational overhead it introduces.

To evaluate above points, the experiments are performed on latent space of LSUN Bedroom images, which are constructed in the same way as Sec. 5.4. Throughout the training process, we record the training time and calculate the FID scores of 10-step sampling at specific epochs. As illustrated in

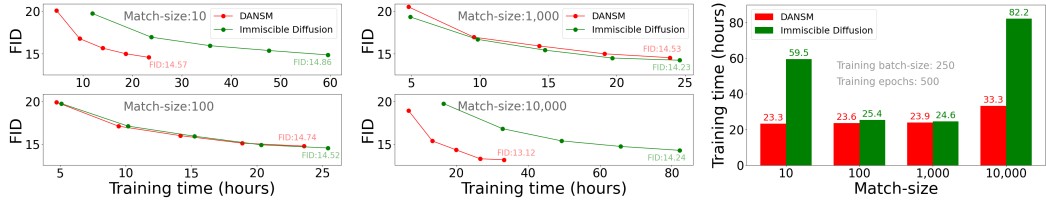

(a) FID of match-size 10 & 100  (b) FID of match-size 1K & 10K  (c) Training time vs. match-size

Figure 7: Comparison of DANSM and Immiscible Diffusion on the latent space of LSUN Bedroom.

Table 2: Path lengths and inter-path distances across different match-sizes ("*ms*"). The "*length*" is path length, and "baseline" refers to random paths of RFM without DANSM. Meanwhile, "$d_{avg}$" and "$d_{min}$" represent the average inter-path distance and minimal inter-path distance, respectively.

| | 50K CIFAR-10 images ($3{\times}32{\times}32$) | | | | | 50K Bedroom latent variables ($4{\times}32{\times}32$) | | | | | | 2K Bedroom ($3{\times}256{\times}256$) | | | |
|---|---|---|---|---|---|---|---|---|---|---|---|---|---|---|---|
| *ms* | baseline | 10 | 100 | 1K | 10K | 100K | baseline | 10 | 100 | 1K | 10K | 100K | baseline | 10 | 100 | 1K |
| *length* | 62.04 | 61.73 | 61.36 | 61.06 | 60.83 | 60.64 | 83.47 | 82.97 | 82.40 | 81.92 | 81.53 | 81.19 | 502.89 | 502.57 | 502.18 | 501.85 |
| $d_{avg}$ | 33.97 | 34.01 | 34.09 | 34.24 | 34.36 | 34.45 | 55.74 | 56.06 | 56.48 | 56.77 | 57.03 | 57.28 | 281.63 | 281.82 | 282.05 | 282.24 |
| $d_{min}$ | 17.99 | 18.02 | 18.06 | 18.10 | 18.13 | 18.17 | 47.65 | 47.88 | 48.17 | 48.41 | 48.62 | 48.79 | 192.43 | 192.46 | 192.54 | 192.64 |

Fig. 6b, the DANSM method significantly reduces training time compared to the baselines. For example, when baseline RFM reaching the FID score of 15.3, the DANSM method with match-size 10,000 cuts training time from 25.5 to 15.4 hours, saving 39.6% of the total training time. It demonstrates that the DANSM method can reduce training time by a large margin.

## 5.6 COMPARISON WITH IMMISCIBLE DIFFUSION

Immiscible Diffusion (Li et al., 2024) has the same goal as DANSM but utilizes the Hungarian algorithm (Kuhn, 1955) as its optimization method. Given $n$ noises and $n$ samples, Immiscible Diffusion incurs a time complexity of $O(n^3)$ in the noise-sample matching procedure while DANSM obtains a more efficient $O(n^2)$ complexity. For fair comparison, two identical RFM models are trained from scratch for 500 epochs on the latent space of LSUN Bedroom by the two methods. Throughout the training process, FID scores are calculated for 10-step sampling results at specific epochs: 100, 200, 300, 400, and 500, and total training time is tracked to analyze the overhead introduced by the noise-sample matching. The dots in Fig. 7 represent the epochs at which FID scores are calculated. As shown in the figure, DANSM achieves comparable FID scores with Immiscible Diffusion but requires less training time, especially with match-size of 10,000. Further analysis is provided in A.4.

## 5.7 ABLATION

Ablation studies are conducted on the only hyperparameter of DANSM — match-size. Tab. 2 shows how the match-size influences the path lengths and inter-path distances (both average and minimal) across three datasets: CIFAR-10, Bedroom latent, and Bedroom image. For CIFAR-10, the path length starts at 62.04 and decreases gradually as match-size increases to 100K, where the path length becomes 60.64. The average inter-path distance ($d_{avg}$) and minimal inter-path distance ($d_{min}$) follow a slightly upward trend, with $d_{avg}$ increasing from 33.97 to 34.45, and $d_{min}$ from 17.99 to 18.17. The similar trends are shown for Bedroom latent and Bedroom image datasets. It should be highlighted that even slight shortening in path length, or slight increasing in inter-path distance, can contribute to speed up the training process significantly.

Furthermore, to better disclose the mechanism of DANSM, we report the $t^*$ timestep (Eq. 4) that DANSM finds. As explained in Sec. 3.1, $t^*$ is the timestep at which the closest inter-path distance occurs. $t = 0$ corresponds to clean sample and $t = 1$ represents pure noise. Meanwhile, different datasets have different data distributions, which leads to different $t^*$ values. For example, the CIFAR-10 image pixel values are scaled to the range $[-1, 1]$, resulting in mean-variance values of $(-0.053, 0.253)$. In contrast, the latent space variables in SD exhibit mean-variance values of $(0.13, 0.68)$. Their $t^*$ values are compared in Tab. 3, which also shows the contrasts of path lengths

Table 3: Data of preferable and random paths on different datasets. The terms "$d_{avg}$" and "$d_{min}$" refer to the average and minimal inter-path distances, respectively, while "$t^*_{avg}$" and "$t^*_{min}$" represent the timesteps where the average and minimal distances occur.

| | 50K CIFAR-10 images ($3{\times}32{\times}32$) | | | | 50K Bedroom latent ($4{\times}32{\times}32$) | | | | 10K SD latent ($4{\times}64{\times}64$) | | | |
|---|---|---|---|---|---|---|---|---|---|---|---|---|
| | length | $d_{avg}$ | $t^*_{avg}$ | $d_{min}$ $t^*_{min}$ | length | $d_{avg}$ | $t^*_{avg}$ | $d_{min}$ $t^*_{min}$ | length | $d_{avg}$ | $t^*_{avg}$ | $d_{min}$ $t^*_{min}$ |
| preferable | 49.88 | 35.50 | 0.1645 | 18.52 0.0075 | 55.10 | 68.76 | 0.2004 | 56.51 0.0100 | 129.27 | 130.83 | 0.2882 | 115.08 0.1559 |
| random | 62.04 | 33.97 | 0.1916 | 17.99 0.0557 | 83.47 | 55.74 | 0.3803 | 47.65 0.2870 | 165.91 | 110.93 | 0.3761 | 98.69 0.2993 |

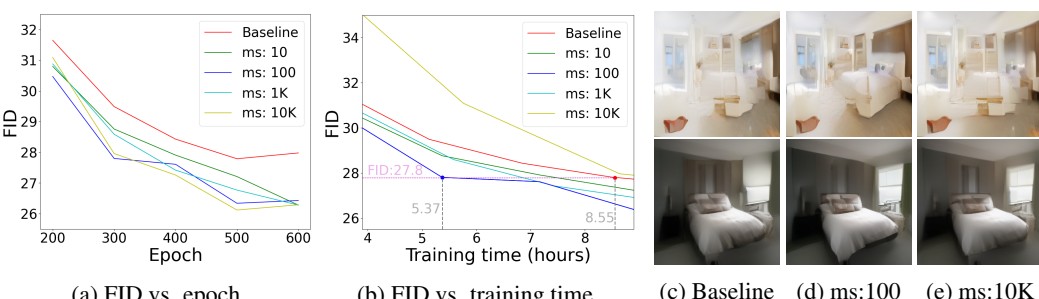

| (a) FID vs. epoch | (b) FID vs. training time | (c) Baseline (d) ms:100 (e) ms:10K |
|---|---|---|

Figure 8: DM on latent of Bedroom by 10-step sampling: comparison between baseline and DANSM with various match-sizes ("*ms*"). (c)-(e) images are generated by models at the same training time (5.37 hours) but with different training methods.

and inter-path distances between preferable and random paths. These results show that the preferable paths have shorter lengths, greater average and minimal inter-path distances, and lower $t^*$ values compared to random paths. Another notable finding is that $t^*$ values are small in both preferable and random paths, indicating the closest inter-path distance happens near the clean sample.

## 5.8 GENERALIZATION TO GENERAL DIFFUSION MODELS

DANSM is proposed based on rectified flow models. However, it can be generalized to diffusion models (DM) (Song et al., 2021a) without any extra adaptation. On CIFAR-10 and latent Bedroom datasets, DANSM outperforms baseline in DM model. Tab. 4 shows that on CIFAR-10, it yields best performance with match-size 1,000, reducing the FID from 93 to 58. Moreover, Fig. 8 shows the improvement on latent Bedroom. To reach FID of 27.8, the baseline requires 8.55 hours whereas DANSM with match-size 100 takes only 5.37 hours (as shown in Fig. 8b), resulting in a 37.2% reduction in training time.

Table 4: FID↓ comparison between different epochs of DM with DANSM.

| DM with DANSM on CIFAR-10 (3-step sampling) | | | | | |
|---|---|---|---|---|---|
| *epoch* | 200 | 400 | 600 | 800 | 1,000 |
| baseline | 120 | 106 | 99 | 94 | 93 |
| ms=1,000 | **89** | **70** | **62** | **59** | **58** |
| ms=5,000 | 142 | 108 | 89 | 79 | 73 |

## 6 CONCLUSION

In this paper, we aim to ease the training process by increasing inter-path distance. Using the straight path property of rectified flow models, we first derive a closed-form formula to calculate the inter-path distance and propose our method, DANSM. Furthermore, we derive the negative correlation between inter-path distance and path length. Based on this relationship, we use path length as a surrogate of DANSM. Although DANSM is developed based on rectified flow models, the experimental results show that it provides excellent speed up for both rectified flow models and diffusion models. DANSM is simple, scalable, and fast, leading to substantial improvements in training efficiency.

### ACKNOWLEDGMENTS

This research is supported by the National Research Foundation, Singapore and Infocomm Media Development Authority under its Trust Tech Funding Initiative. Any opinions, findings and conclusions or recommendations expressed in this material are those of the author(s) and do not reflect the views of National Research Foundation, Singapore and Infocomm Media Development Authority.

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

# A    APPENDIX

## A.1    INTER-PATH DISTANCE DEDUCTION

Given two noise-sample pairs, $(z_1, x_1)$ and $(z_2, x_2)$, the paths are defined by the timestep $t$ as follows:

$$\begin{cases} r_1 = (1-t)x_1 + tz_1 \\ r_2 = (1-t)x_2 + tz_2 \end{cases} \qquad t \in [0,1]. \tag{12}$$

Furthermore, the distance between $r_1$ and $r_2$ is defined as the Euclidean distance between the points corresponding to $t$:

$$\begin{aligned} f_{r_1,r_2}(t) &= \|r_2 - r_1\|_2 \\ &= \|(1-t)x_2 + tz_2 - (1-t)x_1 - tz_1\|_2 \\ &= \|(1-t)(x_2 - x_1) + t(z_2 - z_1)\|_2. \end{aligned} \tag{13}$$

In the following derivation, we simplify the notation by using $f(t)$ to present $f_{r_1,r_2}(t)$. Let $V = x_2 - x_1$ and $U = z_2 - z_1$, we get a new expression of $f(t)$:

$$\begin{aligned} f(t) &= \|(1-t)V + tU\|_2 \\ &= \|V + t(U - V)\|_2. \end{aligned} \tag{14}$$

Its derivative $w.r.t.$ timestep $t$ is:

$$\begin{aligned} f'(t) &= \frac{df(t)}{dt} \\ &= \frac{\left(V + t(U-V)\right)^\top (U-V)}{\|(1-t)V + tU\|_2}. \end{aligned} \tag{15}$$

As $f(t)$ is a concave parabola, its minimal value occurs when the derivative $f'(\hat{t}) = 0$.

$$\hat{t} = \frac{V^\top (V-U)}{(V-U)^\top (V-U)}. \tag{16}$$

## A.2    DEDUCTION OF EQ. 9

Using the law of cosine, we derive $cos\,\gamma$ as:

$$\begin{aligned} cos\,\gamma &= \frac{\|U\|^2 + \|V\|^2 - \|U-V\|^2}{2\|U\| \cdot \|V\|} \\ &= \frac{2UV}{2\|U\| \cdot \|V\|} \\ &= \frac{2(z_2 - z_1)(x_2 - x_1)}{2\|U\| \cdot \|V\|} \\ &= \frac{2z_2 x_2 + 2z_1 x_1 - 2z_2 x_1 - 2z_1 x_2}{2\|U\| \cdot \|V\|} \\ &= \frac{x_1^2 + z_1^2 + x_2^2 + z_2^2 - 2z_2 x_1 - 2z_1 x_2 - (x_1^2 + z_1^2 + x_2^2 + z_2^2 - 2z_2 x_2 - 2z_1 x_1)}{2\|U\| \cdot \|V\|} \\ &= \frac{\|x_2 - z_1\|^2 + \|x_1 - z_2\|^2 - (\|x_1 - z_1\|^2 + \|x_2 - z_2\|^2)}{2\|U\| \cdot \|V\|} \\ &= \frac{\|q_1\|^2 + \|q_2\|^2 - (\|p_1\|^2 + \|p_2\|^2)}{2\|U\| \cdot \|V\|}. \end{aligned} \tag{17}$$

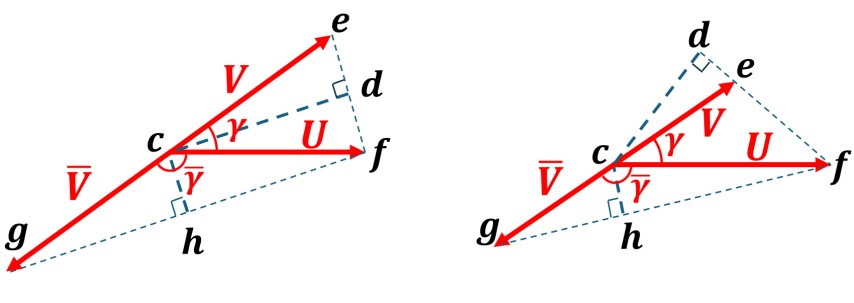

(a) Point $d$ locates in line segment $\overline{ef}$          (b) Point $d$ locates out of line segment $\overline{ef}$

Figure 9: Comparing the inter-path distances between case 1 and case 2 of Fig. 3.

## A.3  Cosine Value vs. Inter-Path Distance

In this section, we compare the inter-path distances of case 1 and case 2 from Fig. 3 of Sec. 4.1. The basic setup is illustrated in Fig. 9a. Case 1 involves the triangle $\triangle cfe$, where $\angle fce = \gamma$, $\overline{cd} \perp \overline{ef}$ and point $d$ is the foot of the perpendicular. Similarly, case 2 involves the triangle $\triangle cfg$, where $\angle fcg = \overline{\gamma}$, $\overline{ch} \perp \overline{fg}$ and point $h$ is the foot of the perpendicular. Given that the vectors $V = -\overline{V}$, we have $\|V\| = \|\overline{V}\|$, and both $V$ and $\overline{V}$ lie on the same straight line. Consequently, the two triangles have equal areas, $\triangle cfe = \triangle cfg$. On the other hand, as concluded in Sec. 4.1, $cos\,\overline{\gamma} < 0$ and $cos\,\gamma > 0$. Therefore, the angles $\overline{\gamma} > \gamma$. All these variables are shown in Fig. 9. The lengths of the line segments have the following relationship:

$$
\begin{cases} \triangle cfg = \triangle cfe \\ \overline{\gamma} > \gamma \end{cases}
$$
$$
\Rightarrow \begin{cases} \|c - h\| \cdot \|g - f\| = \|c - d\| \cdot \|e - f\| \\ \|g - f\| > \|e - f\| \end{cases} \tag{18}
$$
$$
\Rightarrow \|c - h\| < \|c - d\|.
$$

Meanwhile, since $\gamma$ is an acute angle, the perpendicular foot $d$ may occupy different positions relative to the line segment $\overline{ef}$. It could lie on $\overline{ef}$, as shown in Fig. 9a, or fall outside of $\overline{ef}$, as illustrated in Fig. 9b. In the latter scenario, the inter-path distance is $\|c - e\|$. Notably, $\|c - h\| < \|c - d\| < \|c - e\|$, thus confirming that the inter-path distance in case 1 is greater than that in case 2.

## A.4  Comparison with Immiscible Diffusion

The key feature of Immiscible Diffusion (Li et al., 2024) is the noise-sample assignment by "one line of code", which stems from Scipy (Virtanen et al., 2020) library. The code snippet is:

```
scipy.optimize.linear_sum_assignment()
```

However, the above code operates only at the CPU level and is incompatible with GPU acceleration. This limitation prevents it from leveraging parallel processing techniques, making it unsuitable for efficiently handling large-scale inputs. Therefore, as the match-size increases, such as 10,000 in Fig. 7c, the computational overhead of noise-sample assignment (referred to as *overhead*) rises significantly, becoming a major bottle neck of the training process. Unfortunately, the Immiscible Diffusion paper evaluates their generation quality solely based on the number of training epochs (as shown in their figures 4, 6, and 8), without considering the overhead. Our DANSM method takes computational overhead into account, ensuring a thorough and fair evaluation by comparing generation quality based on total training time, including the overhead (as in Figs. 6b, 7, and 8b).

It is worth noting that in Fig. 7c, Immiscible Diffusion requires much more time than DANSM with match-size of 10. The reason is closely related to the batch size used in the experiment, which is set to 250. Consequently, for each batch, Immiscible Diffusion performs 25 CPU-based operations

to optimize the noise-sample matching. These repeated CPU calls become the primary source of overhead, impacting the overall computation efficiency.

Additionally, we present FID comparison results based on training epochs. This evaluation excludes the computational overhead of the noise-sample matching procedures and focuses solely on the effects of data optimization. As shown in Fig. 10, both methods exhibit comparable effectiveness in optimizing noise-sample matching.

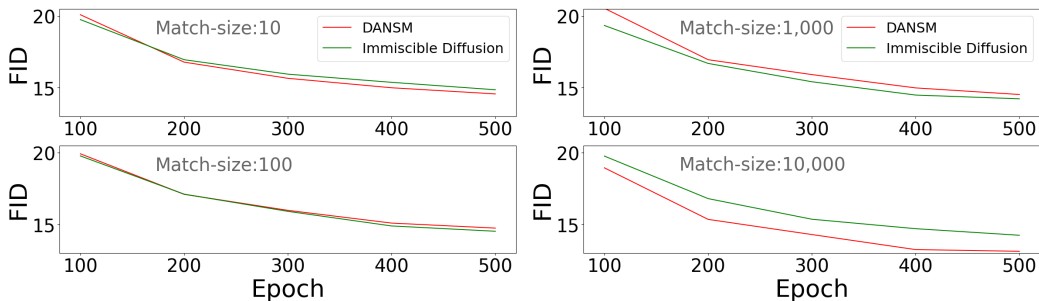

Figure 10: FID↓ comparison between DANSM and Immiscible Diffusion for RFM models on the latent space of the LSUN Bedroom dataset. The FID values are computed based on a 10-step sampling process. Both methods demonstrate similar FID scores across various match-sizes.

A.5    VISUAL COMPARISON OF DIFFERENT METHODS TRAINED ON LATENT OF BEDROOM

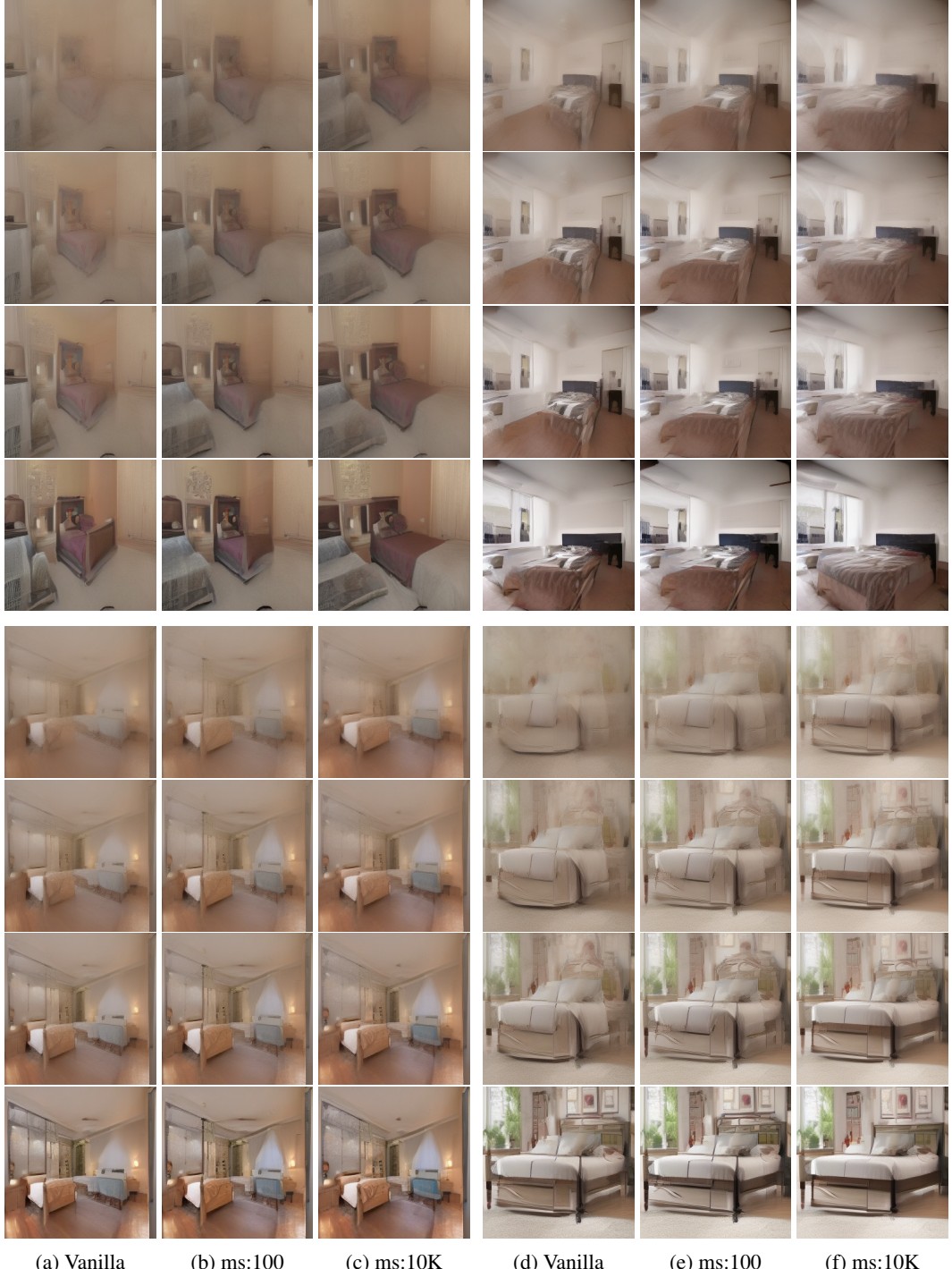

|  (a) Vanilla  |  (b) ms:100  |  (c) ms:10K  |  (d) Vanilla  |  (e) ms:100  |  (f) ms:10K  |

Figure 11: RFM on latent of Bedroom. Visual comparison of images generated by RFM models. The models are trained on the same latent space using different methods: vanilla RFM (columns $a$ and $d$), DANSM with match-size 100 (columns $b$ and $e$), and DANSM with match-size 10K (columns $c$ and $f$). Different rows are generated by different sampling steps: 3-step (rows 1 and 5), 4-step (rows 2 and 6), 5-step (rows 3 and 7), and 10-step (rows 4 and 8).

## A.6 VISUAL COMPARISON OF DIFFERENT METHODS TRAINED ON IMAGENET

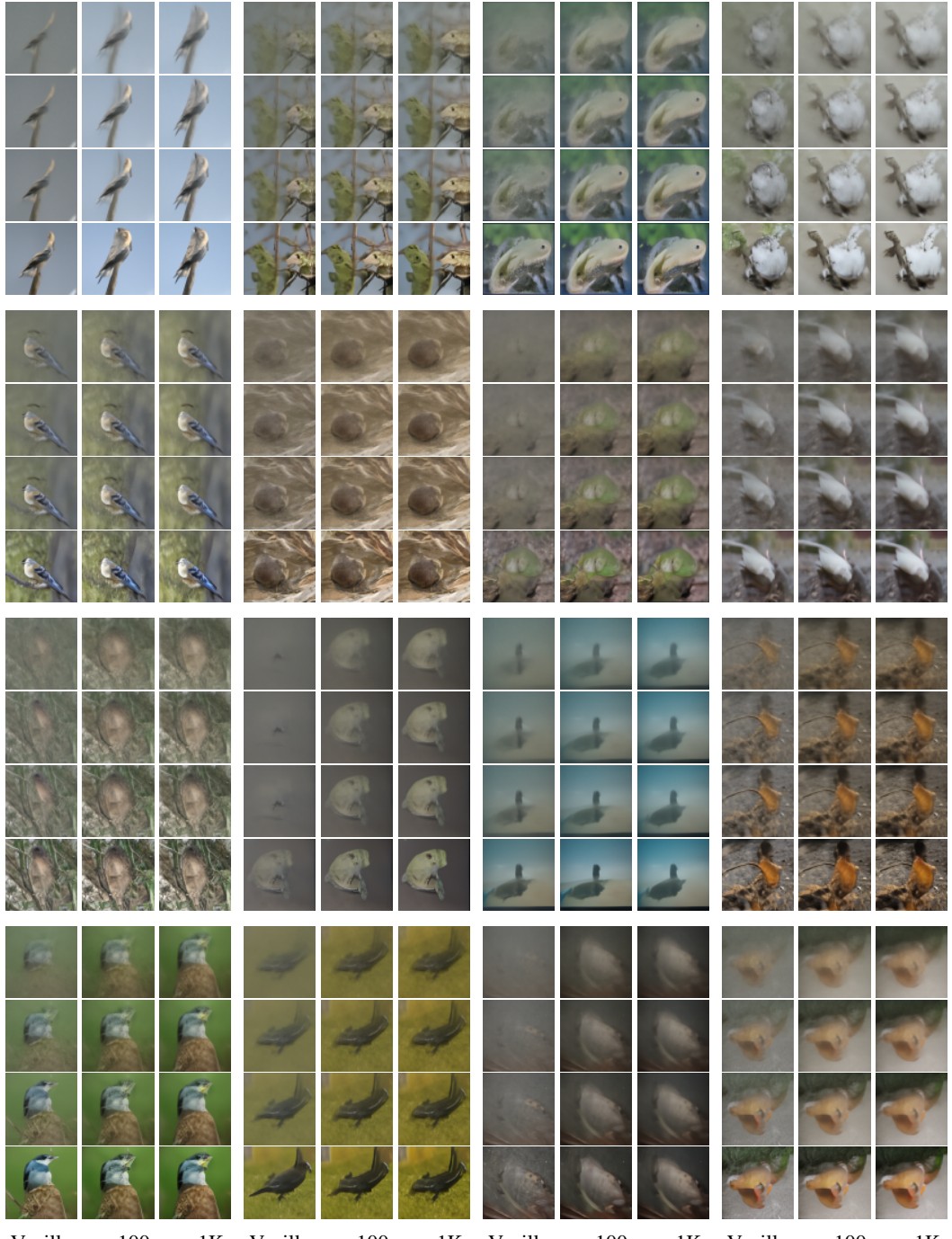

Vanilla  ms:100  ms:1K    Vanilla  ms:100  ms:1K    Vanilla  ms:100  ms:1K    Vanilla  ms:100  ms:1K

Figure 12: Visual comparison of ImageNet images (64×64) generated by RFM models. The models are trained on the same dataset using different methods: vanilla RFM, DANSM with match-size 100 ("*ms:100*"), and DANSM with match-size 1000 ("*ms:1K*"). Different rows are generated by different sampling steps: 3-step (rows 1, 5, 9 and 13), 4-step (rows 2, 6, 10 and 14), 5-step (rows 3, 7, 11 and 15), and 10-step (rows 4, 8, 12 and 16).

## A.7 COMPARISON OF DIFFERENT METHODS TRAINED ON FFHQ

Table 5: FID↓ comparison between different epochs of RFM training processes on FFHQ(256×256) images, where "*ms*" means match-size of DANSM. The FID scores are calculated on images generated by 10-step sampling process.

| epoch | 20 | 40 | 60 | 80 | 100 |
|---|---|---|---|---|---|
| baseline | 121.92 | 104.59 | 97.25 | 89.97 | 86.76 |
| ms=100 | 102.51 | 95.08 | 86.64 | **69.24** | **58.58** |
| ms=1,000 | **95.46** | **87.24** | **78.18** | 75.27 | 66.95 |

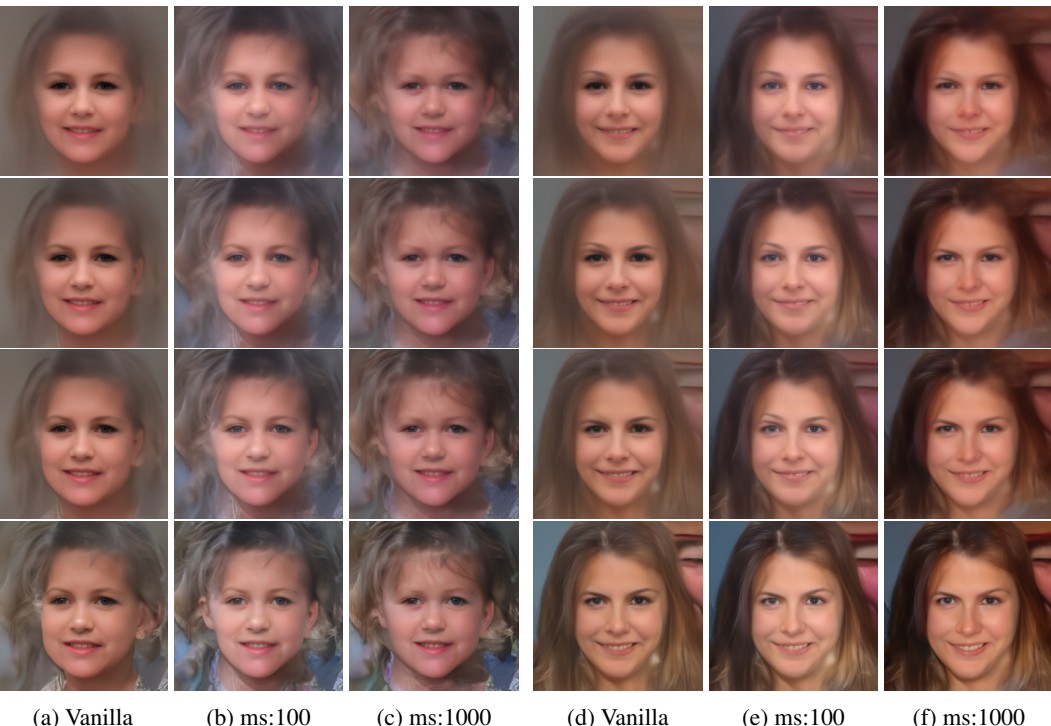

| (a) Vanilla | (b) ms:100 | (c) ms:1000 | (d) Vanilla | (e) ms:100 | (f) ms:1000 |
|---|---|---|---|---|---|

Figure 13: RFM on FFHQ. Visual comparison of images generated by RFM models. The models are trained on the same dataset using different methods: vanilla RFM (columns $a$ and $d$), DANSM with match-size 100 (columns $b$ and $e$), and DANSM with match-size 1000 (columns $c$ and $f$). Different rows are generated by different sampling steps: 3-step (rows 1), 4-step (rows 2), 5-step (rows 3), and 10-step (rows 4).

## A.8 FID COMPARISON WITH FEWER STEPS

Table 6: FID↓ comparison of RFM model training at different epochs and match-sizes ("*ms*").

| | RFM with DANSM on CIFAR-10 (2-step sampling) | | | | | | RFM with DANSM on Bedroom (2-step sampling) | | | | |
|---|---|---|---|---|---|---|---|---|---|---|---|
| epoch | 100 | 200 | 300 | 400 | 500 | epoch | 20 | 40 | 60 | 80 | 100 |
| baseline | 171 | 171 | 170 | 169 | 165 | baseline | 280 | 243 | 215 | 199 | 188 |
| ms=5,000 | 88.13 | 87.91 | 84.61 | 83.67 | 82.41 | ms=100 | 248 | 203 | **184** | 173 | 186 |
| ms=50,000 | **74.42** | **73.42** | **74.15** | **73.53** | **73.52** | ms=500 | **209** | **188** | 186 | **164** | 163 |

