# OpenReview forum: "Easing Training Process of Rectified Flow Models Via Lengthening Inter-Path Distance"
_ICLR.cc/2025/Conference — ICLR 2025 Spotlight_

### Official Review · Reviewer_Ly7e · 2024-10-31

**Soundness:** 3
**Presentation:** 3
**Contribution:** 2
**Rating:** 6
**Confidence:** 4

**Summary:**

The paper reveals that different diffusion models trained on the same dataset tend to produce similar outputs when given the same input noise. This suggests the existence of "preferable noise-sample pairs" in the training process. The authors propose the Distance-Aware Noise-Sample Matching method to lengthen the inter-path distance for speeding up the training of diffusion-based models. The experiments show that the proposed method improves the training speed by about 30%~40%.

**Strengths:**

The paper presents a framework that simplifies the optimization of inter-path distances to path length optimization, based on the observations of noise-sample pair relationships in diffusion models.

The approach leverages closed-form formulas derived from rectified flow models to enable efficient optimization without requiring architectural modifications, making it easy to integrate with existing methods.

**Weaknesses:**

The authors heavily emphasize "consistent model reproducibility," which seems interesting but abruptly transitions to proposing a method for improving training speed. This makes the main idea of the paper quite unclear.

The presentation of the proposed method is vague, and Algorithm 1 cannot be effectively reproduced with the provided steps. I suggest adding mathematical formulations in Sections 3.3 and 4.2 to rigorously explain how the proposed model is optimized.

Regarding experiments, the paper lacks visual comparison results, which are crucial for method evaluation, especially when the FID quantitative metrics show minimal differences at the tiny scale discussed. I also recommend comparing with more methods and incorporating the proposed training speed improvement approach across more SD models. I believe the paper should present higher-resolution and larger-scale generation results. Given the computational benefits brought by this work's methodology, these tasks should be more computationally feasible.

**Questions:**

Why is it necessary to learn latent-noise pairs that are well-aligned in the t-SNE space? This might suggest that straight flow matching is preferable, but the underlying reasoning is not clearly explained in the paper. Furthermore, I am confused about whether "consistent model reproducibility" is beneficial or detrimental for diffusion-based models. The fact that different model architectures can generate similar images from the same noise might indicate limited generation patterns due to the isotropic nature of the Gaussian noise.

---

> ### Author Response · Authors · 2024-11-23
> **Response to reviewer Ly7e (1/2)**
>
> We sincerely appreciate your recognition of our method's simple optimization and easy integration.
> Thank you for your questions, Our detailed responses are given below.
>
> **Weakness 1: The authors heavily emphasize "consistent model reproducibility," which seems interesting but abruptly transitions to proposing a method for improving training speed. This makes the main idea of the paper quite unclear.**
>
> Consistent model reproducibility shows that given the same training images and same input noise, different models generate similar images.
> It motivates us to visualize the relationship between noise and training image as shown in Fig. 2. Consistent model reproducibility and Fig. 2 imply that each training image prefers some noise than others.
> These noises are named preferable noises in the paper.
> More clearly, training images prefer the noises, whose noise-to-image paths bearing longer inter-path distance.
> Since the current training scheme randomly assigns noise to image, without considering the preference of the training images, it slows down the training process. In this paper, based on these observations, we propose to optimize the inter-path distance for speeding up the training process.
> To smooth the transitions and improve the readability, we have revised the main paper and removed the unnecessary emphasis of consistent model reproducibility.  In addition, in section 2.2, we add the statement below to avoid confusion.
>
> > "We would like to highlight that the aim of this paper is not to analyze consistent model reproducibility. However,  it inspires us to maximize the inter-path distance to speed up the training."
>
> **Weakness 2 --- The presentation of the proposed method is vague, and Algorithm 1 cannot be effectively reproduced with the provided steps. I suggest adding mathematical formulations in Sections 3.3 and 4.2 to rigorously explain how the proposed model is optimized.**
>
> Thank you for your suggestions for improving presentation and reproducibility of the work.
> In the revised paper, the mathematical formations have been added in Section 3.3 and 4.2, which are also given below. To ensure the reproducibility of our work, we will share our code to other researchers once this paper is published.
>
> The inter-path distance maximization in Sec. 3.3 is:
> \begin{equation}
>     \max_{\sigma}{\frac{2}{n(n-1)} \sum_{i=1}^n{\sum_{j=i+1}^n{dist(p_{i, \sigma(i)}, p_{j,\sigma(j)})}}} ,
> \end{equation}
> where $\sigma$ is a permutation with input from $1,2,\cdots,n$.
> The term $dist(p_{i, \sigma(i)}, p_{j,\sigma(j)})$ denotes the distance between
> the paths $p_{i,\sigma(i)}$ and $p_{j,\sigma(j)}$, as defined in Sec. 3.1 of the revised paper.
> This formulation optimizes the permutation $\sigma$ to maximize the average inter-path distance among all path pairs.
> The path length minimization in Sec. 4.2 is:
> \begin{equation}
>     \min_{\sigma}{\frac{1}{n} \sum_{i=1}^n{||p_{i,\sigma(i)}||}} .
> \end{equation}
> It has lower computational complexity than the inter-path distance maximization by an order of magnitude.
>
>
> **Weakness 3 --- Regarding experiments, the paper lacks visual comparison results, which are crucial for method evaluation, especially when the FID quantitative metrics show minimal differences at the tiny scale discussed. I also recommend comparing with more methods and incorporating the proposed training speed improvement approach across more SD models. I believe the paper should present higher-resolution and larger-scale generation results. Given the computational benefits brought by this work's methodology, these tasks should be more computationally feasible.**
>
> Thank you very much for your suggestions.
> In the revised paper, we have included visual image comparisons in Fig. 6c, Fig. 8c, and in Appendices A.5, A.6, and A.7 for further analysis and clarity.
> The resolution of the generated LSUN bedroom images and ImageNet images are respectively 256 by 256 and 64 by 64.
> Comparing the baseline, DANSM always generates higher quality images with more detailed features. Although DANSM can speed up the training process more than 37\%, according to our experimental results  (Fig. 6b and Fig. 8b), training SD on large-scale high-resolution images is still beyond our GPU resources. More clearly, training SD requires 200,000 A100 GPU-hours [1].
> ```code
> [1] https://huggingface.co/stabilityai/stable-diffusion-2-1
> ```

---

> ### Author Response · Authors · 2024-11-23
> **Response to reviewer Ly7e (2/2)**
>
> **Question 1 --- Why is it necessary to learn latent-noise pairs that are well-aligned in the t-SNE space? This might suggest that straight flow matching is preferable, but the underlying reasoning is not clearly explained in the paper.**
>
> To understand the property clearly, we have trained a rectified flow model on CIFAR-10. During the training, we collected the losses and corresponding minimal and average inter-path distances of the training samples. Then, we grouped them according to their minimal inter-path distances and average inter-path distances and calculated their mean training losses in each group. Fig. 4(b) and (c) illustrate the relationship between training losses and inter-path distances. They show that samples with shorter minimal or average inter-path distance associate with higher training losses. In other words, they are harder to train.
> We also provided an extreme case, where two paths interest (in Section 3.2), to explain why they are harder to train. These results align with t-SNE plots in Fig. 2. The models prefer paths with longer inter-path distance, which associate with lower training losses.
> Based on the relationship between inter-path distance and path length discussed in Section 4.1,
> longer inter-path distance implies shorter path length, which makes noise-to-sample matching look like straight flow matching, as the reviewer observed.
> These results also explain why DANSM can effectively speed up the training process. It exploits sample-to-noise matching to increase the inter-path distance, which eases the training. The property can also be observed in Fig. 6 and Fig. 8.
> Given the same training time or epoch, DANSM always provides better results, implying that it converges faster than the baseline.
> In the revised paper, we provide the results and the discussion to explain the relationship between training loss and inter-path distance.
>
> **Question 2 --- Furthermore, I am confused about whether "consistent model reproducibility" is beneficial or detrimental for diffusion-based models. The fact that different model architectures can generate similar images from the same noise might indicate limited generation patterns due to the isotropic nature of the Gaussian noise.**
>
> "Consistent model reproducibility” is a property of the diffusion-based models. For this work, it is beneficial since it motivates us to identify the preferable noise and preferable path properties. Based on these properties, we propose DANSM to speed up the training. The aim of this work is to speed up the training process, not to analyse the consistent model reproducibility.
> Although this question is out of the scope of our work, we try to answer it.
> Diffusion-based models try to learn a mapping between noise space and data space. Well-trained models should generate data close to the distribution of training data. Even using different architectures, given the same training data, ideally, the distributions of generated data should be similar. Thus, the generation limit is on the distribution of the training data.
> Some researchers are analysing consistent model reproducibility.
> The reviewer can refer their works [2][3][4]. It should be reemphasized again that analysing  consistent model reproducibility is out of scope of our work.
>
> ```code
> [2] Zhang et al. The emergence of reproducibility and consistency in diffusion models. ICML 2024.
> [3] Liu et al. Flow straight and fast: Learning to generate and transfer data with rectified flow. ICLR 2023
> [4] Song et al. Score-based generative modeling through stochastic differential equations. ICLR 2021
> ```

---

> > ### Comment · Reviewer_Ly7e · 2024-11-27
> >
> > Thank you for your detailed response. The point-by-point clarifications have resolved most of my concerns, so I have decided to raise my rating.

---

> > > ### Author Response · Authors · 2024-11-27
> > >
> > > Dear Reviewer Ly7e,
> > >
> > > Thank you sincerely for your thoughtful feedback and the upgraded score. Based on your valuable comments, we have refined the main paper and will ensure these improvements are reflected in the final version.  Thank you very much once again.
> > >
> > > Best regards,
> > >
> > > The Authors

---

### Official Review · Reviewer_zPez · 2024-11-03

**Soundness:** 3
**Presentation:** 3
**Contribution:** 3
**Rating:** 8
**Confidence:** 2

**Summary:**

This paper explores how different diffusion methods yield similar results with the same dataset, identifying "preferable noises" for samples. It introduces the Distance-Aware Noise-Sample Matching (DANSM) method to optimize training by increasing inter-path distances. DANSM significantly speeds up training by 30%–40% without losing quality, offering insights into enhancing diffusion model efficiency.

**Strengths:**

1. The diagram is very clear, making it easy to understand how different diffusion methods yield similar results with the same dataset, identifying "preferable noises" for samples.

2. The proposed method significantly speeds up training by 30%–40% without losing quality, offering new insights into enhancing diffusion model efficiency.

**Weaknesses:**

1. The FID calculations are measured with very few sampling steps, making it difficult to ensure the quality of the generated results. It would be beneficial to provide a comparison of different methods while maintaining image quality, ideally with visualizations, such as images decoded using stable diffusion.

2. The experiments were only validated on CIFAR and LSUN-BEDROOM datasets. Validation on a broader range of datasets, such as ImageNet and FFHQ, would provide more comprehensive insights. Additionally, including some actual generated images for visual comparison would be advantageous.

**Questions:**

1. Include a comparison using longer sampling steps.

2. Conduct more extensive experiments on complex datasets.

---

> ### Author Response · Authors · 2024-11-23
>
> We sincerely appreciate your recognition of our method's easy understanding and efficiency.
>
> **Weakness 1 --- The FID calculations are measured with very few sampling steps, making it difficult to ensure the quality of the generated results. It would be beneficial to provide a comparison of different methods while maintaining image quality, ideally with visualizations, such as images decoded using stable diffusion.**
>
> **Question 1 --- Include a comparison using longer sampling steps.**
>
> Thank you for your suggestion. We have conducted additional experiments and tested the proposed method on CIFAR-10,
> ImageNet64 and LSUN Bedroom datasets using more sampling steps.
> The new quantitative results are reported in Table 1 in the revised paper, and the original contents of Table 1 are put in Appendix A.8.
> Meanwhile, in Appendix A.5, A.6 and A.7, we also provide more images generated by different sampling steps for visual comparison.
> Fig. 6(c) and Fig. 8(c) show images generated by different models trained on the same amount of time for comparison.
> On training time perspective, we have replotted Fig. 6(b) for more clear comparison.
> Given the same FID, which measures the quality of generated images, the proposed method DANSM uses significantly less training time than the baseline.
> For example, DANSM using match size (ms) of 10K only required 15.4 hours to achieve FID of 15.3 on the latent of LSUN Bedroom dataset but the baseline required 25.5 hours, which is 39.5\% reduction of training time (see Fig. 6b).
> In addition to RFM, we also provide new results using more sampling steps for diffusion model (DM) on the latent space of LSUN Bedroom dataset.
> The results are given in Fig. 8.
> Same as Fig. 6(b), Fig. 8(b) shows that DANSM can effectively reduce the training time given the same FID. More precisely, for FID 27.8, DANSM using match size of 100 only used 5.37 hours but the baseline used 8.55 hours, which is 37\% reduction of training time. Training SD requires 200,000 A100 GPU-hours [1]. Even using DANSM to speed up the training, it is beyond our GPU computation power.
>
> ```code
> [1] https://huggingface.co/stabilityai/stable-diffusion-2-1
> ```
>
> **Weakness 2.1 --- The experiments were only validated on CIFAR and LSUN-BEDROOM datasets. Validation on a broader range of datasets, such as ImageNet and FFHQ, would provide more comprehensive insights.**
>
> **Question 2 --- Conduct more extensive experiments on complex datasets.**
>
> Following your suggestions, we have conducted the experiments on ImageNet and FFHQ. The numerical results of ImageNet are given in Table 1.
> Same as other experiments, they indicate the effectiveness of the proposed method and the benefits of using large match size.
> The visual comparisons of ImageNet are given in Appendix A.6.
> Of FFHQ, the numerical result and corresponding visual comparisons are given in Appendix A.7.
> Meanwhile, the numerical results of the two new datasets are also given below.
>
> **ImageNet of 5-step sampling**
> | epoch | 100 | 200 | 300 | 400 | 500 |
> | -     | -   | -   | -   | -   | -   |
> | baseline | 74.2 | 68.4 | 64.3 | 60.8 | 57.1 |
> | ms=100   | 74.1 | 67.5 | 61.2 | 57.6 | 54.7 |
> | ms=1000  | **72.2** | **64.5** | **58.1** | **54.5** | **52.2** |
>
> **FFHQ of 10-step sampling**
> | epoch | 20 | 40 | 60 | 80 | 100 |
> | -     | -  | -  | -  | -  | -   |
> | baseline | 121.92 | 104.59 | 97.25 | 89.97 | 86.76 |
> | ms=100   | 102.51 |  95.08 | 86.64 | **69.24** | **58.58** |
> | ms=1000  | **95.46**  | **87.24**  | **78.18** | 75.27 | 66.95 |
>
> **Weakness 2.2 --- Additionally, including some actual generated images for visual comparison would be advantageous.**
>
> Thank you for your suggestion about qualitative visual comparisons.
> New generated results of different models and datasets are provided in the revised paper.
>
> - RFM model on latent of LSUN Bedroom: Fig. 6(c) and Appendix A.5;
>
> - DM model on latent of LSUN Bedroom: Fig. 8(c);
>
> - RFM model on ImageNet: Appendix A.6;
>
> - RFM model on FFHQ: Appendix A.7.

---

> ### Comment · Reviewer_zPez · 2024-11-25
>
> Thank you for your response. My concerns have been fully addressed, and I would like to update my evaluation to accept.

---

> > ### Author Response · Authors · 2024-11-26
> >
> > Dear Reviewer zPez
> >
> > We sincerely appreciate your feedback and the upgraded score. Your insightful suggestions have been instrumental in improving the clarity and quality of our manuscript. Based on your valuable advice, our manuscript has improved a lot, and we will keep those changes in the final version. Thank you very much for your support.
> >
> > Best regards,
> >
> > The Authors

---

### Official Review · Reviewer_oVk9 · 2024-11-03

**Soundness:** 4
**Presentation:** 3
**Contribution:** 3
**Rating:** 8
**Confidence:** 2

**Summary:**

This work proposes Distance-AwareNoise-Sample Matching (DANSM) to increase training speed in generative models without any loss (in fact gain) in performance. The proposed method is inspired by "the difference between random paths used in training and preferable paths from well-trained models".
The work is theoretically motivated and sound.
The work offers limited evaluations, however, they seem to be sufficient to make the case.

**Strengths:**

The work is well written except for a few typos (for example, in line 198/199 "To **analysis**" is written instead of "To **analyse**", and in line 309/310 "have not been **evaluation** on RFM" is written instead of **evaluated**).

The idea is sound and is theoretically motivated, the results are in line with the hypothesis.
The implementation seems simple and straightforward.

The claims made in the paper seem to be backed by empirical evaluations.

**Weaknesses:**

I have 2 major points in this regard:
1. The plots are not well made, some of the lines in the plots, for example, Figure 6(b), Figure 7(b), and Figure (b) seem to be starting from arbitrary locations. It would help to invest more time and make better-looking plots without these artifacts, or if they are not artifacts then an explanation for the same would be very helpful.

2. The work lacks comparisons to "Immiscible Diffusion", while the paper does point out the key differences in lines 307-315, it would be interesting to see empirical evaluations in comparison to "Immiscible Diffusion" since the methods are closely related.

**Questions:**

Q1- I would appreciate a better explanation for Figure 6 (b), the plot looks a bit unclear to me. Additionally, there are lines behind the legend curbing my ability to understand the plot completely. My essential question is that unlike Figure 7, in Figure 6 (b) I do not see any gains in training times when using the proposed method. Is this understanding of mine correct?

---

> ### Author Response · Authors · 2024-11-23
>
> We sincerely appreciate your recognition of our paper’s sound idea, simple implementation and well organization!
>
> **Comment 1 --- The work is well written except for a few typos (for example, in line 198/199 "To analysis" is written instead of "to analyse", and in line 309/310 "have not been evaluation on RFM" is written instead of evaluated).**
>
> Thanks for the detailed review and feedback, it's really helpful to remind us to further improve our paper writing quality by strictly checking any possible typos.  During this rebuttal period, we have carefully proofread the full manuscript again to check any possible grammatical mistakes and typos, and corrected those spotted by you such as line 198/199 and 309/310.
>
> **Weakness 1 --- The plots are not well made, some of the lines in the plots, for example, Figure 6(b), Figure 7(b), and Figure (b) seem to be starting from arbitrary locations. It would help to invest more time and make better-looking plots without these artifacts, or if they are not artifacts then an explanation for the same would be very helpful.**
>
> Based on your comments and suggestion, we reconsider the Figure 6 and Figure 7 in original submitted version of the paper. Both Figure 6 and 7 bring similar information about relationship between FID and training time, although they were plotted on different settings. Therefore, we decide to remove Figure 7 from the revised version to make our paper more concise.
> Furthermore, we have replotted Figure 6 to clear the artifacts and make sure the subplots aligned with same settings, and added visualization results in Figure 6(c). Hope the new figure can provide better explanation and showcase to readers.
>
> **Weakness 2 --- The work lacks comparisons to "Immiscible Diffusion", while the paper does point out the key differences in lines 307-315, it would be interesting to see empirical evaluations in comparison to "Immiscible Diffusion" since the methods are closely related.**
>
> Following your suggestion, we have added a new subsection (Sec. 5.6) to discuss and compare with the work "Immiscible Diffusion" comprehensively.
> We performed the experiments on latent of LSUN Bedroom dataset. The FID results and training time are shown in Figure 7 of the revised version of paper, which are highlighted in blue color in the new submission.
>
> The figure demonstrates that while DANSM and Immiscible Diffusion achieve comparable FID scores,
> DANSM requires significantly less training time, highlighting its efficiency advantage.
> With match-size of 10,000 and 500-epoch training, DANSM achieves FID 13.12 vs. 14.24 of Immiscible Diffusion.
> Moreover, to reach such FID performance, DANSM takes 33.3 hours for training while Immiscible takes 82.2 hours --- 246\% training time of DANSM (see Figure 7(c)). In fact, given $n$ noises and $n$ samples, Immiscible Diffusion incurs a time complexity of $O(n^3)$ in the noise-sample matching procedure, and its related code is limited to CPU execution. In contrast, the proposed DANSM method obtains a more efficient $O(n^2)$ complexity and supports GPU acceleration. We have added more analysis on it in Appendix A.4.
>
> **Question 1 --- I would appreciate a better explanation for Figure 6 (b), the plot looks a bit unclear to me. Additionally, there are lines behind the legend curbing my ability to understand the plot completely. My essential question is that unlike Figure 7, in Figure 6 (b) I do not see any gains in training times when using the proposed method. Is this understanding of mine correct?**
>
> We have replotted the Figure 6 (b) to better visualize the performance in the revised paper.
> As shown in the replotted Figure 6 (b), to obtain the same FID (e.g., 15.3), the proposed method with 10,000 match-size used 15.4 hours while baseline method required 25.3 hours, which shows that the proposed method gained 39.27\% saving of training time.

---

> > ### Comment · Reviewer_oVk9 · 2024-11-23
> >
> > Dear Authors
> >
> > Thank you for the responses and the revisions, the manuscript looks better now (at least in my humble opinion).
> >
> > Regarding, "Immiscible Diffusion v/s DANSM",
> > My current understanding is that the respective optimization method used makes the theoretical difference in the time complexity, the hardware used is secondary, and the "speed up" in model training due to DANSM is due to theoretical reasons, and not practical implementation reasons. If that is indeed the case, then my suggestion would be to avoid discussing the CPU optimization steps v/s GPU optimization in the main paper, this is something that has been discussed in the appendix and is also better suited for the appendix as it is a low-level implementation detail. There currently is no GPU-acceleration support for the scipy operations, but these are simple mathematical operations that can be easily written in CUDA to achieve GPU speed-ups.
> >
> > Additionally, I believe a better way to discuss the time difference would be plotting FID v/s wall clock time instead of the number of epochs, to show the gain in speed, either in the main paper or the supplementary.
> >
> > Lastly, is there a hypothesis as to why only for match size 1000, DANSM performs worse than Immiscible Diffusion but outperforms it (or matches its performance for match size=100) in the other scenarios?
> >
> > Best Regards
> >
> > Reviewer oVK9

---

> ### Author Response · Authors · 2024-11-24
>
> Dear reviewer oVK9,
>
> Thank you for your prompt response.
>
> You are correct regarding the ''Immiscible Diffusion v/s DANSM'' comparison.
> The key differences lie in the theoretical foundation,
> particularly the deduction of the negative correlation between inter-path distance and path length,
> as well as the computational complexity ($O(n^3)$ from Immiscible Diffusion  v/s $O(n^2)$ from DANSM).
>
> We discussed the CPU v/s GPU hardware issues to highlight the significant disparity in training times.
> However, as you pointed out, this analysis is not central to the main discussion of the paper.
> A more appropriate place for such details is the appendix.
> Following your suggestion, we have moved the CPU-GPU discussion to appendix A.4.
>
> Additionally, we have re-plotted the figures to show a comparison of FID scores v/s wall clock time,
> which is now reflected in the updated Fig. 7.
> The original FID v/s epoch figures have been moved to appendix A.4 for further reference.
>
> For match size of 1000, Immiscible Diffusion and DANSM have FID scores of 14.23 and 14.53, respectively.
> The two scores are very close to each other, indicating the similar effectiveness of the two methods.
> Given the same number of training epochs, we expect that Immiscible Diffusion and DANSM
> perform similar as shown in Fig. 10 in the appendix of the revised paper, since both methods optimize for the same objective.
> In the four comparisons, Immiscible Diffusion slightly outperforms DANSM for match size of 100 and 1000
> but DANSM outperforms it on the other two match sizes.
> However, it should be highlighted that DANSM uses significantly less training time than Immiscible Diffusion (see Fig. 7).
>
> Best Regards
>
> Authors

---

> > ### Comment · Reviewer_oVk9 · 2024-11-25
> >
> > Dear Authors,
> >
> > Thank you very much for the answer and for including the suggestions In my current understanding, the paper looks good, and I recommend accepting it.
> >
> > Best Regards
> >
> > Reviewer oVk9

---

> ### Author Response · Authors · 2024-11-26
>
> Dear Reviewer oVk9,
>
> Thank you sincerely for your thoughtful feedback and the upgraded score. Following your suggestions and comments, we have improved the main paper a lot and will keep those changes in the final version. Thank you very much again.
>
> Best regards,
>
> The Authors

---

### Meta-Review · Area_Chair_HQuh · 2024-12-15

**Metareview:**

The paper presents a novel method for optimizing diffusion models, demonstrating significant training speed improvements (30%-40%) without compromising quality, and provides theoretical support backed by empirical evaluations. The approach offers a practical solution for accelerating training in diffusion models while maintaining high performance. The use of closed-form formulas for optimizing inter-path distances is both efficient and easy to integrate into existing architectures. However, prior to the rebuttal, the reviewers pointed out that the presentation could be improved.

After the rebuttal, all reviewers recognized the contributions of the paper, with feedback generally leaning towards acceptance. The AC thoroughly reviewed the paper and rebuttal, agreeing with the consensus recommendation for acceptance based on the consistency of the feedback.

**Additional Comments On Reviewer Discussion:**

The reviewers noted that most of the concerns raised have been addressed, leading to a unanimous recommendation for acceptance.

---

### Decision · Program_Chairs · 2025-01-22

Accept (Spotlight)